# A Novel Method for the Generation of Realistic Lung Nodules Visualized Under X-Ray Imaging

**Ahmet Peker** [1,2,*,†] **, Ayushi Sinha** [3,†] **, Robert M. King** [1] **, Jeffrey Minnaard** [4] **, William van der Sterren** [4] **, Torre Bydlon** [3] **, Alexander A. Bankier** [1] **and Matthew J. Gounis** [1]

1   Department of Radiology, University of Massachusetts Medical Center, Worcester, MA 01655, USA; robert.king@umassmed.edu (R.M.K.); alexander.bankier@umassmemorial.org (A.A.B.); matthew.gounis@umassmed.edu (M.J.G.)
2   Department of Radiology, Koc University Hospital, Istanbul 34010, Turkey
3   Philips Research North America, Cambridge, MA 02141, USA; ayushi.sinha@philips.com (A.S.); torre.bydlon@philips.com (T.B.)
4   Philips Image Guided Therapy Systems, 5684 PC Best, The Netherlands; jeffrey.minnaard@philips.com (J.M.); william.van.der.sterren@philips.com (W.v.d.S.)
*   Correspondence: apeker@kuh.ku.edu.tr
†   These authors contributed equally to this work.

**Abstract:** Objective: Image-guided diagnosis and treatment of lung lesions is an active area of research. With the growing number of solutions proposed, there is also a growing need to establish a standard for the evaluation of these solutions. Thus, realistic phantom and preclinical environments must be established. Realistic study environments must include implanted lung nodules that are morphologically similar to real lung lesions under X-ray imaging. Methods: Various materials were injected into a phantom swine lung to evaluate the similarity to real lung lesions in size, location, density, and grayscale intensities in X-ray imaging. A combination of *n*-butyl cyanoacrylate (n-BCA) and ethiodized oil displayed radiopacity that was most similar to real lung lesions, and various injection techniques were evaluated to ensure easy implantation and to generate features mimicking malignant lesions. Results: The techniques used generated implanted nodules with properties mimicking solid nodules with features including pleural extensions and spiculations, which are typically present in malignant lesions. Using only n-BCA, implanted nodules mimicking ground glass opacity were also generated. These results are condensed into a set of recommendations that prescribe the materials and techniques that should be used to reproduce these nodules. Conclusions: Generated recommendations on the use of n-BCA and ethiodized oil can help establish a standard for the evaluation of new image-guided solutions and refinement of algorithms in phantom and animal studies with realistic nodules.

**Keywords:** lung cancer; solitary pulmonary lesions; pulmonary neoplasm; synthetic lung nodules; solid nodules; ground glass opacity; X-ray imaging

## 1. Introduction

Lung cancer is the leading cause of cancer death worldwide, and early diagnosis is crucial for a better prognosis [1]. Consequently, early screening programs have been launched, and along with improved CT imaging, they have led to an increase in the detection of smaller, more peripheral, and both solid and subsolid lesions [2,3]. As a result, more than an estimated 1.5 million people are diagnosed with a pulmonary lesion in the United States within a year [4]. Management of these lesions is challenging for various reasons, including difficulty in accessing small and peripheral lesions, risk of complications like pneumothorax, and challenges in achieving expected diagnostic yield or treatment results using available solutions.

Consequently, safe and accurate diagnosis and treatment of lung lesions is an active area of research, and several solutions have been proposed toward improved image-guided

navigation, transthoracic and endobronchial biopsies, image-guided ablation therapies, and robotic solutions for diagnosis and therapy [5–8]. These solutions must be evaluated in realistic phantom and preclinical studies before they can be sufficiently de-risked to bring to clinical studies. In order to ensure a realistic study environment, implanted lesions, or nodules, must be morphologically similar to real lung lesions in various imaging modalities, including in size, location, density, and grayscale intensities. Radiopaque fiducial markers like metallic beads and coils that have been used in the past do not provide such a realistic study environment as they are much easier to visualize in X-ray imaging than real lung lesions, create metal artifacts, especially in 3D X-ray imaging such as computed tomography (CT) and cone beam CT (CBCT) [9], and are difficult to implant in live animals [10].

As an example of image guidance in endobronchial biopsy procedures, several navigation solutions that are being developed rely on a co-registration between intraoperative fluoroscopy images and a preoperative CT image [11,12]. This co-registration is necessary because small lung lesions, which are visible in CT imaging, are typically not visible under fluoroscopy. Co-registering the two images allows interventionalists to ensure their tools are being navigated toward the lesion that is visible in CT imaging. However, radiopaque fiducial markers are easily visible in fluoroscopy and, therefore, make the co-registration of fluoroscopy and preoperative CT images easier than it would be in real practice, thereby introducing potential bias into the evaluation of the navigation solutions. Realistic lung nodules implanted in phantom or animal models not only enable evaluation in realistic settings, allowing for a better understanding of co-registration algorithm performance in the presence of real lesions, but also enable the curation of richer datasets with ground truth co-registration, which is not available when relying on data with real lesions only, and development of more robust co-registration algorithms. In another example of image guidance, once navigation to the lesion is complete and a biopsy device is deployed, interventionalists often perform a tool-in-lesion confirmation by acquiring a CBCT image using a fixed gantry C-arm system capable of performing a CBCT spin [13]. Emerging image reconstruction techniques have also made such a confirmation possible via a limited sweep tomosynthesis image acquisition using mobile C-arm systems. 3D imaging of metal implants, however, is prone to artifacts such as streaking and blurring due to beam hardening, photon starvation, and other X-ray attenuation characteristics generated by high-density materials [9]. These artifacts can obscure the visualization of features in the field of view and can impair the accurate evaluation of tool position with respect to the metallic fiducial marker. Realistic lung nodules should not generate such artifacts and, therefore, enable improved evaluation of tool position with respect to the nodules. Further, evaluation of tool-in-lesion confirmation is challenging in the presence of metal implants since biopsy devices cannot be inserted into metallic fiducial markers. Realistic lung nodules, however, should allow biopsy devices to be inserted, enabling tool-in-lesion confirmation. Such confirmation may be used to evaluate the accuracy of CT fluoroscopy co-registration algorithms. Finally, metallic fiducial markers are difficult to implant into phantom or preclinical models, especially at peripheral segments of the lung where navigation technologies must be evaluated [10]. Marker placement can require invasive procedures, which can increase the chance of inducing pneumothorax [10,14]. Nodules generated with non-metallic implants that may be inserted into the lung using narrow-gauge needles allow for greater flexibility in nodule placement without inducing pneumothorax, which in turn enables an easier setup for evaluation of X-ray guided endobronchial biopsy techniques, which typically do not induce pneumothorax.

Various techniques have been explored in the past to create more easily implantable materials, including a mixture of petroleum gel and Vaseline, a fat–wax–lipiodol mixture, a gelatin–agar–iodinated contrast mixture, and a calcium hydroxylapatite filler [15–18]. These methods suffer from various disadvantages, resulting in a lack of a standardized process for realistic lung nodule creation. Further, there have been no methods described so far that are capable of creating nodules that mimic ground glass opacity. *n*-butyl cyanoacrylate (n-BCA) is a well-known and widely used liquid embolic agent in interventional radiology and is approved for the embolization of brain arteriovenous malformations. n-BCA consists of monomers that form polymers with adhesive bonds to soft tissues in contact with anions.

Ethiodized oil is iodinated poppy seed oil used as a radiopaque contrast agent, often in conjunction with n-BCA, to aid visibility in X-ray imaging [19].

In this study, we describe a new method that uses n-BCA to create materials that, when implanted into lung tissue, mimic real lung lesions under X-ray imaging, including fluoroscopy, CBCT, and tomosynthesis. These materials can be used to create both solid and subsolid nodules, including ground glass opacity, and are easily implantable. Through different injection techniques, these materials can also be used to create nodules of different shapes. These nodules can subsequently be used in future studies to evaluate image-guided solutions that aim to aid the diagnosis and treatment of lung lesions.

## 2. Materials and Methods

### 2.1. Ex Vivo Swine Lungs

In order to evaluate materials for their similarity to real lung lesions, we performed a preliminary set of experiments on ex vivo swine lungs. Swine lungs were used because of the similarities between swine and human airway and lung anatomy [20]. Two different sets of preserved swine lungs were used (NASCO, Fort Atkinson, WI, USA); no animals were used solely for this purpose. The lungs (Figure 1A) were inserted into a custom-made vacuum chamber that was connected to a pump (Figure 1B). The vacuum chamber was equipped with multiple holes for inserting instruments, like percutaneous needles. The holes were occluded with silicone stoppers to maintain negative pressure in the chamber. The lungs were inflated and then injected with various materials. For storage and preservation between multiple imaging sessions over several months, the lungs were stored in humectant (70% propylene glycol and 30% water).

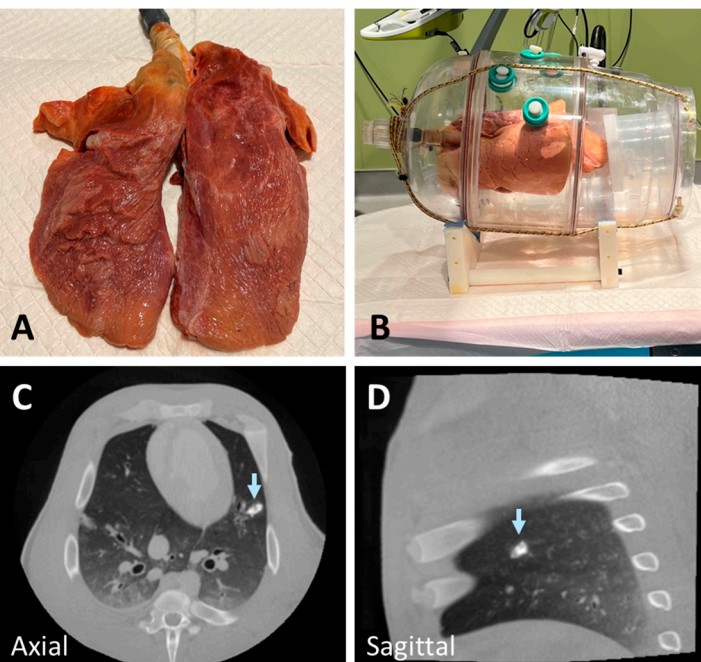

**Figure 1.** Deflated swine lung (**A**) inserted into a custom-made vacuum chamber and connected to a vacuum pump (**B**). The lung was inflated (**B**) using the vacuum pump to generate negative pressure in the chamber before materials were injected. An example of a nodule created using a mixture of n-BCA and hydroxyapatite showing higher radiopacity than lung lesions (arrow) (**C**,**D**).

### 2.2. Post-Mortem Swine Lungs

We also evaluated materials in post-mortem swine models. Two post-mortem swine, four lungs, were used. The swine was used in studies unrelated to lungs, which were approved by the Institutional Animal Care and Use Committee (IACUC) before being

euthanized. No animals were used solely for this purpose. Euthanized swine remained intubated and ventilated via a mechanical ventilator for the injection of synthetic nodules.

### 2.3. Lung Nodule Preparation and Injection

In the initial experiments to develop realistic lung lesion-mimicking nodules, we tested various materials and combinations, including Ethylene–Vinyl Alcohol Copolymer (Onyx), agar solution, wax, gelatin, and n-BCA. To enhance radiopacity, we incorporated different proportions of barium, tantalum powder, hydroxyapatite, and ethiodized oil. Onyx, despite its potential, did not solidify effectively after injection into tissue and exhibited excessively high radiopacity due to its tantalum powder content. Similarly, a mixture of n-BCA and hydroxyapatite resulted in high radiopacity, which was not representative of realistic lung nodules (Figure 1C,D). Agar and gelatin required heating before injection and solidified within approximately 30 min. However, their hardness decreased at body temperature, making them less suitable for live implantation.

Wax presented additional challenges, as it needed to be heated to high temperatures before injection but rapidly solidified upon cooling to room temperature. This limited the time available for precise injection, further compounded by its high viscosity. In contrast, a combination of n-BCA and ethiodized oil demonstrated radiopacity closely resembling lung lesions while being more practical to handle. These materials required no heating prior to use and maintained a consistency that allowed for smooth injection, making them the most suitable choice for our study.

For solid nodules, n-BCA and ethiodized oil (TRUFILL n-BCA, Cordis Neurovascular Inc., Miami, FL, USA) were mixed at a ratio of 1 g n-BCA to 10 mcg ethiodized oil. For ground glass nodules, 1 g of n-BCA was used. Injections were performed percutaneously via 19 G needles while the lungs were inflated. Materials were injected slowly and continuously over 10 s to allow enough time for n-BCA polymerization. To create spiculated margins resembling malignant lesions, half of the material was injected slowly within the first 5 s, followed by a rapid injection of the remaining material to produce the desired effect. A 5 mL syringe was used, and needles were flushed with 5% dextrose solution before injection to prevent premature polymerization of n-BCA. A small amount of material was injected into the needle tract to prevent pneumothorax while the needle was being retracted from the lung. All injections were performed by a radiologist with experience in over 100 image-guided lung biopsy procedures.

### 2.4. Imaging Protocol

The CBCT scans were performed within 30 min after the agent injection. This time frame allowed us to ensure there were no complications, properly position the vacuum chamber or swine, and implement the required radiation safety measures. CBCT scans were performed with a Philips Allura FD20 (Philips, Amsterdam, The Netherlands) X-ray imaging system using a dedicated lung imaging protocol. The protocol consists of an 8 s roll scan acquiring 480 projection images in a 180-degree sweep at a tube voltage of 120 kV, a pulse width of 10 ms, and a source-to-detector distance of 119.5 cm, generating a $384 \times 384 \times 297$ pixel volume. Saline was placed into the chamber to compare the absorption coefficient or Hounsfield units (HU) in images generated by the system. Additionally, the projection sequence used to reconstruct the CBCT images was used to reconstruct limited sweep tomosynthesis images. Projection sequences spanning $-50$ to $50$ degrees around the anteroposterior (AP) direction were used to reconstruct tomosynthesis images from a 100-degree sweep. Fluoroscopy images were also acquired at AP.

## 3. Results

### 3.1. Ex Vivo Swine Lungs

A solid nodule implanted using our protocol is seen in CBCT and tomosynthesis images in Figure 2A–H. The solid nodule was implanted in the right inferior lobe, and the slow release of the n-BCA–ethiodized oil mix in the injection tract during retraction resulted in artifacts resembling pleural extension.

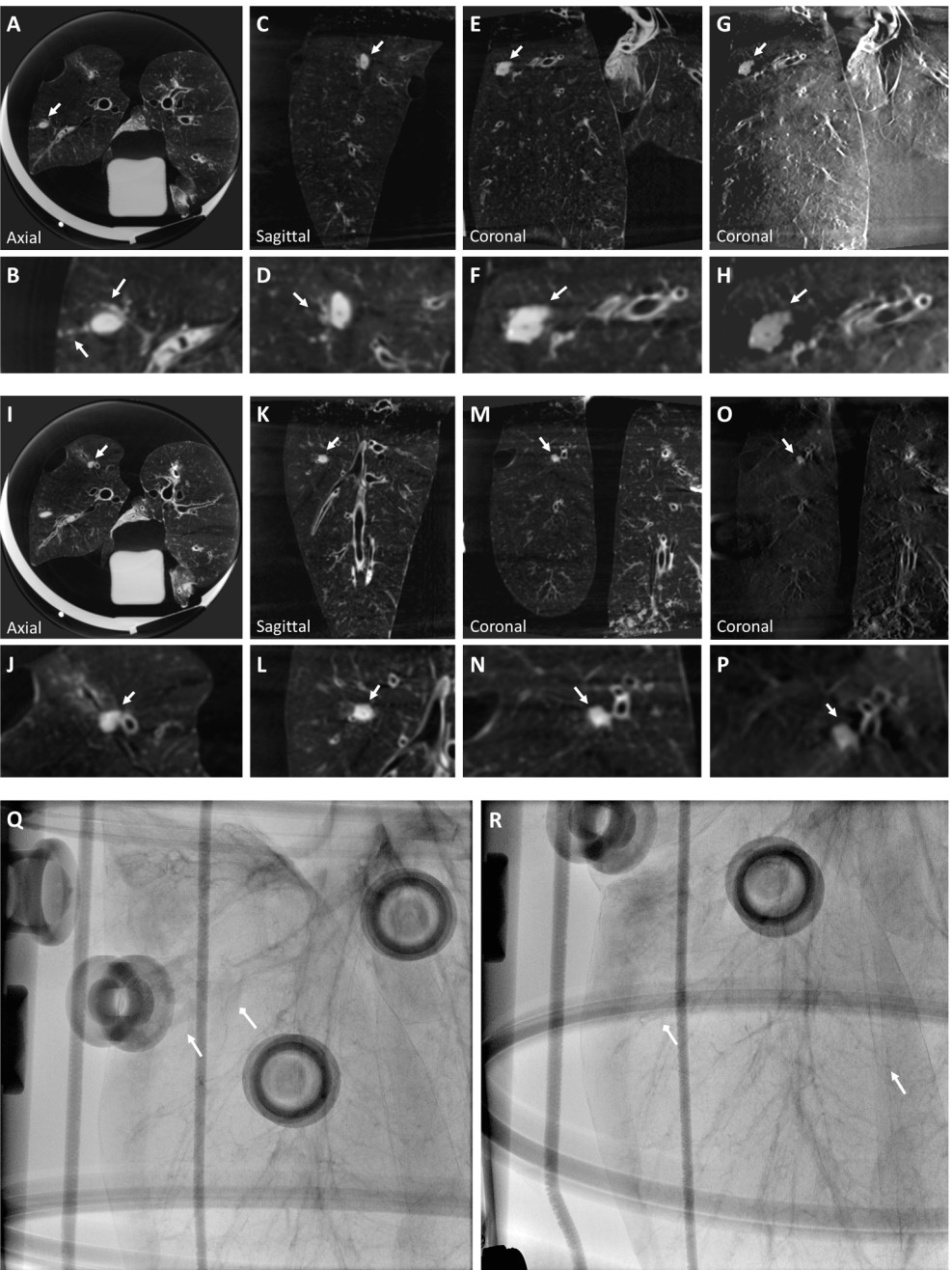

**Figure 2.** A solid nodule (Nodule 1) is visualized in a CBCT image of the right inferior lobe superior segment (**A–F**) of the ex vivo swine lungs. The nodule includes artifacts due to tract injection, which mimic extension from the solid nodule to pleura (**A and inset B**), and satellite micronodules observed in the anterior neighborhood of the nodule (insets **B,D**). The nodule is also visible clearly in tomosynthesis imaging (**G,H**). However, smaller details like the pleural extension are faint (inset **H**). Solid nodule diameter: 11 × 6.5 mm, HU: 32 (±26). A ground glass nodule (Nodule 2) is also visualized in a CBCT image of the right inferior lobe anterior basal segment (**I–N**). The nodule has ill-defined margins extending to the segment bronchus. The nodule is also clearly visible in tomosynthesis imaging (**O,P**), but both anteroposterior (**Q**) and lateral (**R**) fluoroscopy views of the right lobe show neither the solid nodule nor the ground glass nodule, which reflects the properties of real lung nodules. Arrows indicate where the solid (triangular arrow) and ground glass (diamond arrow) nodules are approximately located (**Q,R**). Ground glass nodule diameter: 8 × 7 mm; HU: −360 (±55). Arrows indicate all described nodules.

A ground glass nodule is seen in CBCT and tomosynthesis images in Figure 2I–P. Ground glass opacity (GGO) was generated using n-BCA without ethiodized oil. The nodule, implanted in the right inferior lobe anterior basal segment, shows ill-defined margins extending to the segment bronchus. Neither the solid nor the ground glass nodules are visible in a fluoroscopy image of the lung (Figure 2Q,R). Injecting the materials with a 19 G needle did not result in pneumothorax.

### 3.2. Post-Mortem Swine Lungs

A solid nodule implanted in the right superior lobe anterior segment is shown in CBCT and tomosynthesis images in Figure 3. This nodule is a pleural-based lobulated nodule and depicts irregular margins. These lobulated features may be caused by the injected mixture entering into several nearby alveoli. An expert user may be able to replicate these features by making small perturbations in the needle tip position during injection.

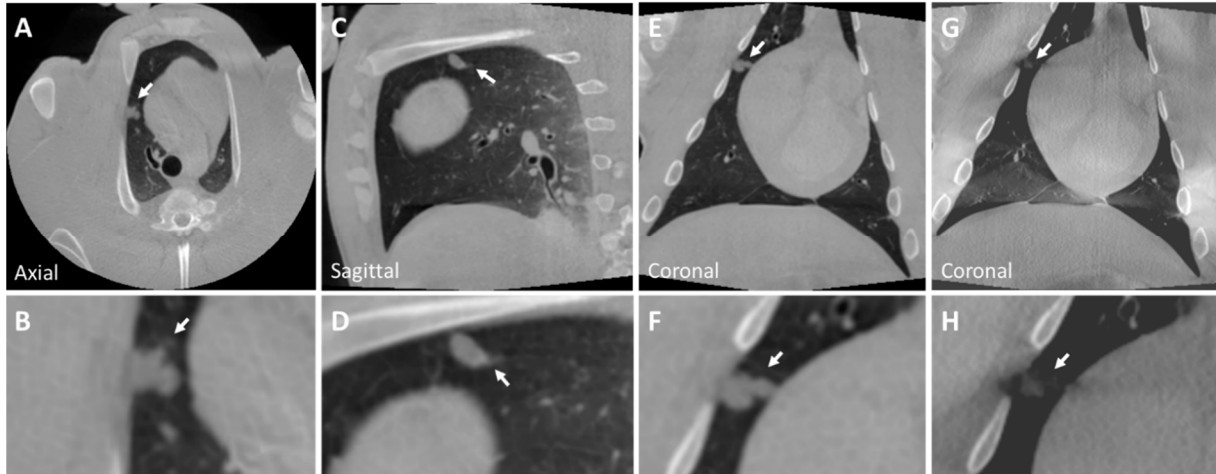

**Figure 3.** CBCT image of a solid nodule (Nodule 3) in the right superior lobe anterior segment (**A**–**F**) of the post-mortem swine lungs. This nodule is a pleural-based lobulated nodule and depicts irregular margins. The nodule is lightly visible in tomosynthesis imaging (**G**,**H**). However, the lobulated characteristics of the nodule are not captured in tomosynthesis imaging (inset **H**). Solid nodule diameter: $10 \times 9$ mm; HU: 25 ($\pm$20). Arrows indicate all described nodules.

A ground glass nodule implanted in the left inferior lobe is shown in CBCT and tomosynthesis in Figure 4. The internal structure of the nodule is heterogeneous and has ill-defined borders with spiculated margins. The spiculated margins result from the adhesive properties of n-BCA, which adheres to surrounding tissue, making its borders appear irregular.

Additionally, a nodule mimicking a malignant lesion created by rapid injection is seen in CBCT and tomosynthesis images in Figure 5. This nodule, implanted in the lower lobe of the right lung, depicts a heterogeneous internal structure and has irregular spiculations mimicking malignant lesions with irregular margins. Due to rapid injection, the concentration per unit area of ethiodized oil decreased. Consequently, the X-ray attenuation of the nodule also decreased as the injected material was spread out over a wider area. To overcome this, the concentration of ethiodized oil can be increased slightly to generate nodules mimicking malignant lesions with striations and within the intensity range of real malignant lesions. No pneumothorax was observed. The diameters and attenuations of the generated and real nodules are shown in Table 1.

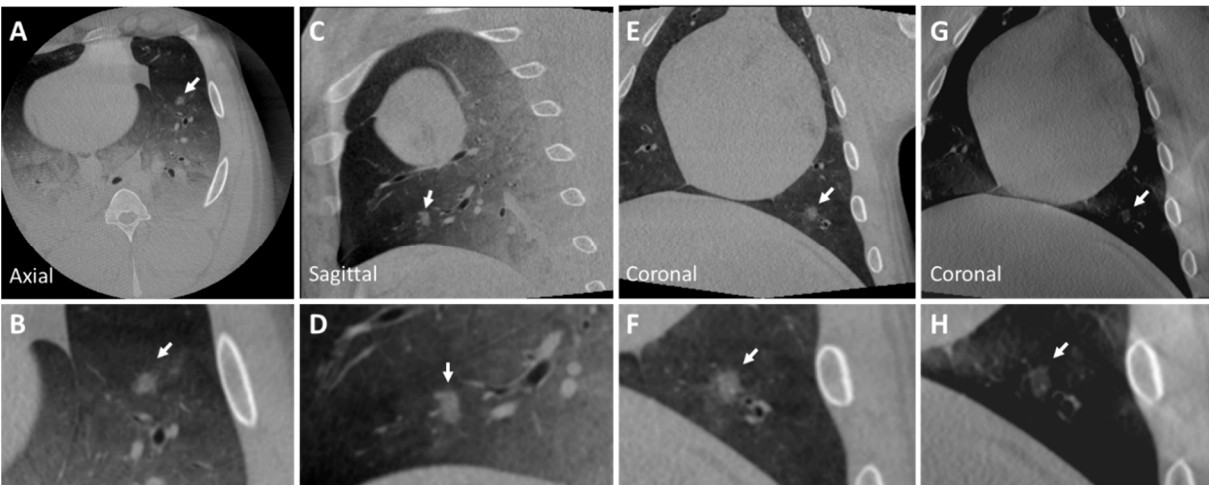

**Figure 4.** CBCT image of a ground glass nodule (Nodule 4) in the left inferior lobe (**A–F**) of the post-mortem swine lungs. The internal structure of the nodule is heterogeneous and has ill-defined borders with spiculated margins (insets **B**,**F**). The nodule is lightly visible in tomosynthesis imaging (**G**,**H**). Nodule diameters: $9 \times 8$ mm; HU: $-330$ ($\pm 50$). The arrow indicates the described nodule.

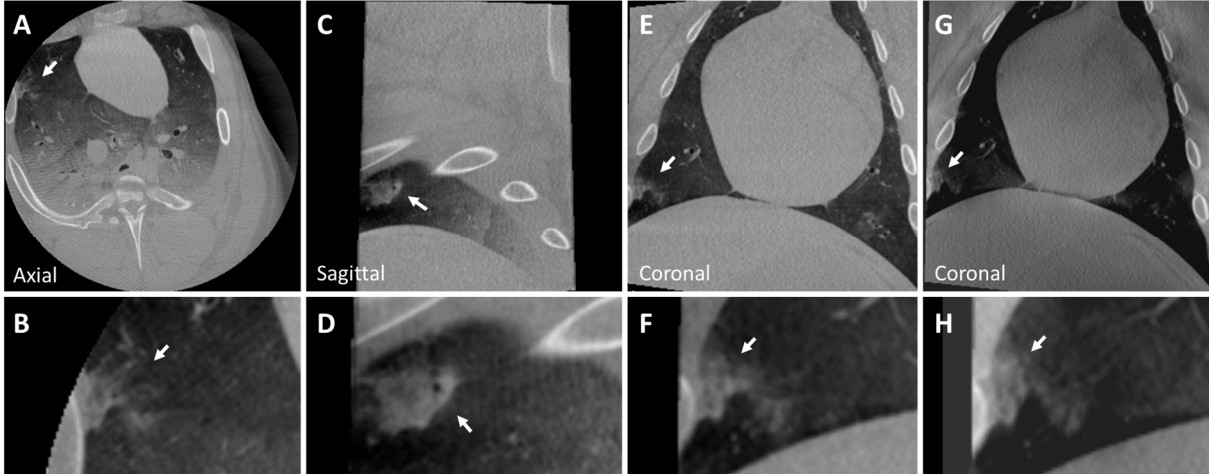

**Figure 5.** CBCT image of a peripherally located malignant nodule (Nodule 5) in the lower lobe of the right lung (**A–G**) of the post-mortem swine lungs (arrows). The internal structure of the nodule is heterogeneous, and it has irregular striations mimicking malignant lesions with irregular margins (inset **B**). While the nodule is visible in tomosynthesis imaging (**G**,**H**), details like striations are not clearly visible (arrows). Nodule diameters: $16 \times 12$ mm; HU: $-310$ ($\pm 40$).

**Table 1.** Diameters and attenuations of real and generated nodules.

|  | Diameter (mm) | CT Attenuation (HU) |
|---|---|---|
| Lung Nodule | <30 | 50 ($\pm 30$) |
| Ground Glass Nodule | <30 | $-200$–$-500$ |
| Nodule 1 | $11 \times 6.5$ | 32 ($\pm 26$) |
| Nodule 2 | $8 \times 7$ | $-360$ ($\pm 55$) |
| Nodule 3 | $10 \times 9$ | 25 ($\pm 20$) |
| Nodule 4 | $9 \times 8$ | $-330$ ($\pm 50$) |
| Nodule 5 | $16 \times 12$ | $-310$ ($\pm 40$) |

*3.3. Recommendations*

➢ Pre-procedure preparation:
- Inflate the phantom lungs before injecting materials.
- Flush needles with a 5% dextrose solution to prevent premature polymerization of n-BCA.

➢ Mixing the Material:
- Manual mixing is sufficient to create a realistic lesion that is not fully homogeneous.
- For a homogeneous lesion, thoroughly mix the n-BCA and ethiodized oil using a shaker.

➢ Mixture for Solid Nodule:
- Use 1 g of n-BCA and 10 mcg of ethiodized oil.

➢ Mixture for Ground Glass Nodule:
- Use 1 g of n-BCA.

➢ Creating the Nodule:
- Inject slowly and steadily, completing the injection within 10 s.

➢ Creating Pleural Extension:
- Inject half of the material slowly and steadily for over 5 s.
- Inject the remaining half slowly while retracting the needle, completing the injection over another 5 s until reaching the pleura.

➢ Obtaining a Lobulated Contour Nodule:
- Move the needle tip in very small forward–backward and up–down motions (no more than 5 mm) during the injection.

➢ Creating Spiculation (Malignant Nodule Characteristic):
- Inject half of the material slowly for over 5 s.
- Inject the remaining portion rapidly in a single burst.

➢ Preventing Pneumothorax:
- Retract the needle while injecting a small amount of material (approximately 0.1 cc) until the needle detaches from the pleura.

## 4. Discussion

In this study, we describe a novel method to create synthetic lung nodules that mimic solid lung lesions, ground glass opacities, and malignant features in CBCT, tomosynthesis, and fluoroscopy images using n-BCA and ethiodized oil. Although CT imaging provides better image resolution and is typically used for screening, our demonstration of realistic lung nodule implantation and visualization in CBCT and tomosynthesis imaging is critical since these imaging modalities are quickly becoming the standard of care for endobronchial and transbronchial biopsies [13]. Synthetic lung nodules such as those presented here can also help in the further development and evaluation of new image-guided solutions, imaging techniques, and image reconstruction methods by refining algorithms in phantom and animal studies with realistic nodules before testing on subjects with real lung lesions. While there have been several earlier attempts at developing lesion-mimicking markers, to the best of our knowledge, prior work does not provide both realistic imaging features as well as ease of nodule implantation.

Silvestri et al. used fiducial markers to create realistic targets to evaluate a bronchoscopic transparenchymal nodule access technique [14]. Fiducial markers are metallic and radiopaque and, therefore, much easier to visualize in X-ray images than real lung lesions. Nelson et al. used a petroleum gel mixed with Vaseline to mimic lung lesions to test the accuracy of scanning-beam digital X-ray tomosynthesis for lung biopsies. This material must be melted before injection, making it difficult to control and implant easily. Additionally,

the absorption coefficient of the generated nodule was around −100 HU [15], which does not reflect the absorption coefficient of real lesions. Absorption coefficients for lung lesions are typically in the range of 50 (±30) HU, with coefficients for benign lesions typically lower than those for malignant lesions [2,21]). Bolte et al. used a fat–wax–ethiodized oil mix to create lung nodules in order to evaluate computer-aided volumetry [16]. While the nodules had an absorption coefficient of approximately 50–150 HU, which is similar to the expected range for real lung lesions, a high-pressure pump was needed to inject this mixture, making implantation challenging [16]. Chen et al. used a solution consisting of bovine skin gelatin, agar, and iodinated contrast injected transbronchially in human cadavers. This solution was boiled for mixing and maintained at 45–50 Celsius for injection, again making implantation challenging [18]. Sterman et al. used hydroxyapatite dermal filler to create pulmonary nodules, which were injected transbronchially into canine models but had difficulty reaching most peripheral regions [17].

Our technique has several advantages over those previously described. First, the absorption coefficient of solid lesions generated using our protocol is similar to that of real lung lesions [21,22], which was evaluated by an expert radiologist. Absorption coefficients can also be adjusted by changing the proportion of ethiodized oil, which can enable the generation of both benign and malignant lesions [21,22]. Second, malignant features such as the extension to the pleura, satellite micronodules, and striations can be imitated as described above. Third, our solution offers a simpler method of injecting/implanting the lesions as it does not require the materials to be heated, melted, or injected under high pressure. Finally, the use of n-BCA and the recommended slow retraction of the needle along with tract injection ensured that we did not cause a pneumothorax in either the ex vivo or post-mortem swine lungs. The development of a pneumothorax, especially in animal experiments, impacts the image quality of the lesion and complicates the procedure, which could lead to early termination of the experiment.

One key limitation of this study is the small sample size used in the animal model, which may affect the robustness and generalizability of the findings. Previous research has shown that small sample sizes in preclinical studies can lead to variability in results and reduce the confidence in conclusions drawn from such experiments. For instance, in studies involving animal models of ischemic stroke, promising results from small cohorts often failed to replicate in larger studies or clinical trials, highlighting the challenges of translating preclinical data into clinical success [23,24]. Therefore, larger-scale studies are necessary to validate the findings and improve the generalizability of these methods.

Another limitation of our study is that all lesion implantations were performed by the same radiologist. Although n-BCA injection has been used in interventional radiology for several years, there is a learning curve in its use, and it may be beneficial to document any variance in the quality of lesions implanted across several users.

## 5. Conclusions

We describe a novel method using n-BCA and ethiodized oil to create solid and ground glass nodules that mimic real lung lesions as well as lesions with malignant features in CBCT, tomosynthesis, and fluoroscopy images. These materials are easy to work with and easy to inject into the lung and do not cause complications like pneumothorax, making our method simple and reproducible for future studies that require synthetic lung nodules.

**Author Contributions:** Conceptualization, A.P., A.S., M.J.G., R.M.K., T.B. and A.A.B.; methodology, A.P., M.J.G., R.M.K., J.M. and A.S.; software, A.S., W.v.d.S. and J.M.; validation, A.P., A.A.B. and R.M.K.; formal analysis, A.S., T.B., W.v.d.S. and J.M.; investigation, A.P., A.S. and R.M.K.; resources, A.S., R.M.K., T.B. and M.J.G.; data curation, J.M., A.S. and W.v.d.S.; writing—original draft preparation, A.P., R.M.K., J.M. and A.S.; writing—review and editing, A.A.B., M.J.G., W.v.d.S. and T.B.; visualization, A.S., J.M. and W.v.d.S.; supervision, A.A.B., T.B. and M.J.G.; project administration, T.B. and M.J.G.; funding acquisition, T.B. and M.J.G. All authors have read and agreed to the published version of the manuscript.

**Funding:** "This research was funded by Philips (United States), grant number PJ-011665", and "The APC was funded by UMASS NECSTR".

**Institutional Review Board Statement:** The swine was used in studies unrelated to lungs, which were approved by the Institutional Animal Care and Use Committee (IACUC) before being euthanized. No animals were used solely for this purpose. The ethical approval is waived.

**Informed Consent Statement:** Not applicable.

**Data Availability Statement:** The data presented in this study are available on request from the corresponding author due to project privacy reasons.

**Conflicts of Interest:** Author Ayushi Sinha, Jeffrey Minnaard, William van der Sterren, and Torre Bydlon were employed by the company Philips. Matthew J. Gounis has research support from Philips Healthcare. The remaining authors declare that the research was conducted in the absence of any commercial or financial relationships that could be construed as a potential conflict of interest. The authors also declare that this study received funding from Philips (United States). The funder was not involved in the study design, collection, analysis, interpretation of data, the writing of this article or the decision to submit it for publication.

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
