# Peer review of "A Novel Method for the Generation of Realistic Lung Nodules Visualized Under X-Ray Imaging"

_tomography, doi:10.3390/tomography10120142_

Round 1

Reviewer 1 Report

Comments and Suggestions for Authors

This review accompanies, “A novel method for the generation of realistic lung nodules visualized under X-ray imaging” by Peker, et al. Overall, this is a good methods paper that describes a new technique for creating implanted lung nodules. However, this paper would be more compelling to the reader if some changes were made to the discussion regarding the comparison of the phantom to physiologically relevant parameters. My comments regarding this manuscript are as follows:

1.      In the abstract, the authors claim to evaluate similarity to real lung lesions in size, location, density, and grayscale intensities in X-ray imaging. However, it would be more compelling to the reader if the nodule diameter and HU were compared to common physiological ranges in a table that summarizes the findings (instead of having to compare discussion text back to various image captions).

2.      In section 2.2, the authors describe using postmortem swine models, but they do not say how many swine were used for this study.

3.      In lines 272-276, the authors state that a benefit of their technique was that no pneumothorax was created. Was this a problem if using other techniques? Or is this just an observation?

Author Response

Thank you very much for taking the time to review this manuscript. Please find the detailed responses below and the corresponding revisions/corrections highlighted changes in the re-submitted files.

Comment 1. In the abstract, the authors claim to evaluate similarity to real lung lesions in size, location, density, and grayscale intensities in X-ray imaging. However, it would be more compelling to the reader if the nodule diameter and HU were compared to common physiological ranges in a table that summarizes the findings (instead of having to compare discussion text back to various image captions).

Response 1. Thank you for your valuable feedback. In line with your suggestion, we have created a table summarizing the nodule diameter and Hounsfield Unit (HU) values, comparing them to common physiological ranges. This table has been incorporated into the Results section to provide a clearer and more concise presentation of the findings (page 9 line 248).

Comment 2.    In section 2.2, the authors describe using postmortem swine models, but they do not say how many swine were used for this study.

Response 2. Thank you for your comment.  In our study, we used two sets of ex vivo swine lungs and two post-mortem swine (four lungs). We added it to the relevant section 2.2.(page 4 line 132-133)

Comment 3.      In lines 272-276, the authors state that a benefit of their technique was that no pneumothorax was created. Was this a problem if using other techniques? Or is this just an observation?

Response 3. Thank you for your comment. Pneumothorax is the most common complication in percutaneous thoracic interventions. Its reported prevalence was 8-64% in lung biopsies and 33-67% in fiducial placement (ref. 10). It is also a problem in experimental settings due to the change in anatomy and nodule position, as well as respiratory instability if a live animal is used. So avoiding pneumothorax is an important advantage of our described technique in lung interventions. Avoiding pneumothorax allows us to create experimental setups that more closely resemble realistic settings during X-ray-guided endobronchial biopsies, which typically do not induce pneumothorax. A clarifying sentence has been added to the introduction (page 2, lines 87-91).

Reviewer 2 Report

Comments and Suggestions for Authors

This study describes a method to generate lung nodules visualized in x-ray imaging. After reviewing the manuscript, I believe it lacks sufficient innovation for publication. The authors utilized n-BCA to mimic lung lesions, which is not a novel approach. Furthermore, the study does not address key aspects of x-ray imaging, such as x-ray energy or beam quality, which are critical factors in imaging performance. The most significant limitation, however, is the lack of clarity regarding the purpose of this method. The authors do not explain why generating realistic lung nodules in a phantom for x-ray imaging is necessary or how this technique would be applied.

Comments on the Quality of English Language

no comment.

Author Response

Thank you very much for taking the time to review this manuscript. Please find the detailed responses below and the corresponding revisions/corrections highlighted changes in the re-submitted files.

Summary. This study describes a method to generate lung nodules visualized in x-ray imaging. After reviewing the manuscript, I believe it lacks sufficient innovation for publication. The authors utilized n-BCA to mimic lung lesions, which is not a novel approach. Furthermore, the study does not address key aspects of x-ray imaging, such as x-ray energy or beam quality, which are critical factors in imaging performance. The most significant limitation, however, is the lack of clarity regarding the purpose of this method. The authors do not explain why generating realistic lung nodules in a phantom for X-ray imaging is necessary or how this technique would be applied.

Response. We thank the reviewer for their feedback. We have added additional details relating to the imaging protocol used (page 5, lines 173-175), and added specific examples detailing how the ability to generate realistic lung nodules would be beneficial in developing and evaluating techniques and algorithms that aid X-ray guided lung biopsy (page 2, lines 63-68, 78-84, 87-91).

Reviewer 3 Report

Comments and Suggestions for Authors

Nice work and well presented. Congratulations to the authors for the work. I have some short questions and a couple of comments:

- In point 2.3, the election of the contrast agent is a bit confusing. I would rephrase the whole first paragraph and divide it in two at least. It's difficult to follow the reasons for selecting the agent.

 - Can you clarify the injecting time required for the n-BCA polymerization? The effects of this time are reflected on the imaging quality, so it should be described in detail.

- In point 2.4 Can you tell the reader how long it takes between the agent injection and the CBCT scans?

- In point 3.3 I would make a list of bullet points for the recommended protocol. That would be easier to read and understand.

My main doubt about your work is why you just made one sample for each model, knowing that this is not a clinical work so you should be free to inject the agents several times in each model for getting a trustable number of images and conclusions. You are assuming the injection protocol affects the images in the same way the agent by itself. For supporting these conclusions you should make, at least, 4-5 injections in each model and compare the images of each procedure, especially because there are some parameters hard to control, like injection rate or bolus pressure. And I would use a gold standard agent to support your conclusions about the benefits of your new approach. Literally, you are saying your agent and protocol are ideal for getting these phantoms. Ideal compared to what?

I strongly recommend increasing the number of injections and images and compare them to a standardized protocol and agent.

Author Response

Summary. Nice work and well presented. Congratulations to the authors for the work. I have some short questions and a couple of comments:

Response. Thank you very much for taking the time to review this manuscript. Please find the detailed responses below and the corresponding revisions/corrections highlighted changes in the re-submitted files.

Comment 1. In point 2.3, the election of the contrast agent is a bit confusing. I would rephrase the whole first paragraph and divide it in two at least. It's difficult to follow the reasons for selecting the agent.

Response 1. Thank you for your feedback. We have carefully rephrased and reorganized the first paragraph in Section 2.3 to improve clarity and readability. The revised version is now divided into two paragraphs, clearly outlining the reasons for selecting the contrast agent. We believe this adjustment addresses your concerns and enhances the overall flow of the section.

 Comment 2. Can you clarify the injecting time required for the n-BCA polymerization? The effects of this time are reflected on the imaging quality, so it should be described in detail.

Response 2. Thank you for your valuable comment. The polymerization of n-BCA occurs almost immediately upon contact with tissue fluids, requiring careful handling and precise injection within seconds to ensure optimal distribution. This rapid polymerization, which is known to vary depending on the n-BCA mixture ratio, typically takes between 0.2 and 5 seconds. However, in lung tissue, where fluid content is lower, this duration is approximately 10 seconds. We have added a detailed explanation of this process to the relevant section to address your concern (pages 5 158-163). Also, we added a detailed guide of our protocol in section 3.3. Recommendations.

Comments 3. In point 2.4 Can you tell the reader how long it takes between the agent injection and the CBCT scans?

Response 3. Thank you for your comment. The CBCT scans were performed within 30 minutes after the agent injection. This time frame allowed us to ensure there were no complications, properly position the vacuum chamber or swine, and implement the required radiation safety measures. We have added this clarification to the relevant section(page 5 lines 168-170).

Comments 4. In point 3.3 I would make a list of bullet points for the recommended protocol. That would be easier to read and understand.

Response 4. Thank you for your comment. We have revised the recommendations section as you suggested. To make our technique easier to follow, we’ve replaced the Table with a more detailed, step-by-step explanation.

  • Pre-procedure preparation:
  • Inflate the phantom lungs before injecting materials.
  • Flush needles with a 5% dextrose solution to prevent premature polymerization of n-BCA.
  • Mixing the Material:
  • Manual mixing is sufficient to create a realistic lesion that is not fully homogeneous.
  • For a homogeneous lesion, thoroughly mix the n-BCA and ethiodized oil using a shaker.
  • Mixture for Solid Nodule:
  • Use 1 g of n-BCA and 10 mcg of ethiodized oil.
  • Mixture for Ground Glass Nodule:
  • Use 1 g of n-BCA.
  • Creating the Nodule:
  • Inject slowly and steadily, completing the injection within 10 seconds.
  • Creating Pleural Extension:
  • Inject half of the material slowly and steadily for over 5 seconds.
  • Inject the remaining half slowly while retracting the needle, completing the injection over another 5 seconds until reaching the pleura.
  • Obtaining a Lobulated Contour Nodule:
  • Move the needle tip in very small forward-backward and up-down motions (no more than 5 mm) during the injection.
  • Creating Spiculation (Malignant Nodule Characteristic):
  • Inject half of the material slowly over 5 seconds.
  • Inject the remaining portion rapidly in a single burst.
  • Preventing Pneumothorax:
  • Retract the needle while injecting a small amount of material (approximately 0.1 cc) until the needle detaches from the pleura.

Comment 5. My main doubt about your work is why you just made one sample for each model, knowing that this is not a clinical work so you should be free to inject the agents several times in each model for getting a trustable number of images and conclusions. You are assuming the injection protocol affects the images in the same way the agent by itself. For supporting these conclusions you should make, at least, 4-5 injections in each model and compare the images of each procedure, especially because there are some parameters hard to control, like injection rate or bolus pressure. And I would use a gold standard agent to support your conclusions about the benefits of your new approach. Literally, you are saying your agent and protocol are ideal for getting these phantoms. Ideal compared to what?

Response 5. I strongly recommend increasing the number of injections and images and compare them to a standardized protocol and agent.

Thank you for your valuable feedback, we completely understand your point. In fact, this study was not conducted on just one model. As mentioned in Section 2.3 (Lung Nodule Preparation and Injection), all the materials we used were tested in different models at various times prior to this study. Based on the results of these preliminary experiments, we developed the technique we used in this study. We then standardized this technique and applied it to two sets of ex vivo swine lungs, and later to two postmortem swine lungs.

To make our approach clearer and more easily applicable, we have updated the recommendations section with more detailed, step-by-step instructions.

We truly appreciate your suggestion regarding increasing the number of injections and comparing them to a standardized protocol. However, we regret to inform you that this is unfortunately not feasible at this point. The project has been completed and is no longer funded. Moreover, the principal investigator, A.P., who was leading this work, has since moved to a different country and is no longer part of the same laboratory. We sincerely apologize for any inconvenience this may cause and appreciate your understanding.

Round 2

Reviewer 2 Report

Comments and Suggestions for Authors

I am alright with the explanations from the authors as per my concerns in this work.

Comments on the Quality of English Language

No comment.

Author Response

Comment. I am alright with the explanations from the authors as per my concerns in this work.

Response. Thank you very much for your positive feedback. We are pleased that the revisions we made in response to your previous comments have addressed your concerns and that you are satisfied with the adjustments. Your acknowledgment of these improvements encourages us, and we are grateful for your valuable input, which has contributed significantly to refining the manuscript.

Reviewer 3 Report

Comments and Suggestions for Authors

Thanks for accommodating my comments and following my recommendations. You made a nice work with that.

I'm still worried about the sample size of this project. Knowing that you cannot make more test in this experiment, I strongly recommend to include a clear reference of that in the discussion and limitations part of the manuscript. Something citing that the short number of test could be affecting the conclusions of the protocol and the confidence of the work.

Author Response

Comment. Thanks for accommodating my comments and following my recommendations. You made a nice work with that.

I'm still worried about the sample size of this project. Knowing that you cannot make more test in this experiment, I strongly recommend to include a clear reference of that in the discussion and limitations part of the manuscript. Something citing that the short number of test could be affecting the conclusions of the protocol and the confidence of the work.

Response. 

We would like to sincerely thank you for your insightful comments and suggestions, which have greatly contributed to the improvement of our manuscript. Your feedback has helped us refine the discussion, making the manuscript more scientifically rigorous, clearer, and ultimately more reproducible. The revisions we made in response to your recommendations have undoubtedly enhanced the overall quality of the work.

Thank you again for your valuable feedback. We have addressed your concerns regarding the small sample size in this study by acknowledging this limitation more clearly in both the Discussion and Limitations sections.

As you suggested, we have included references to studies that highlight how small sample sizes can influence the generalizability and reliability of preclinical findings. Specifically, research in stroke models has shown that promising results obtained from small animal cohorts often fail to replicate in larger trials or clinical settings, leading to concerns about the translation of preclinical results into human applications (23,24). We also emphasize that larger-scale studies are essential for validating our findings and ensuring their broader applicability.

‘’One key limitation of this study is the small sample size used in the animal model, which may affect the robustness and generalizability of the findings. Previous research has shown that small sample sizes in preclinical studies can lead to variability in results and reduce the confidence in conclusions drawn from such experiments. For instance, in studies involving animal models of ischemic stroke, promising results from small cohorts often failed to replicate in larger studies or clinical trials, highlighting the challenges of translating preclinical data into clinical success [23,24]. Therefore, larger-scale studies are necessary to validate the findings and improve the generalizability of these methods.’’

  1. Usui T, Macleod MR, McCann SK, Senior AM, Nakagawa S (2021) Meta-analysis of variation suggests that embracing variability improves both replicability and generalizability in preclinical research. PLoS Biol 19(5): e3001009.
  2.  Ko MJ, Lim CY. General considerations for sample size estimation in animal study. Korean J Anesthesiol. 2021;74(1):23-29.